# Environmental and Genetic Determinants of Serum 25(OH)-Vitamin D Levels during Pregnancy and Early Childhood

**DOI:** 10.3390/children6100116

**Published:** 2019-10-21

**Authors:** Ann-Marie Malby Schoos, Cecilie Vinther, Sarah Nørgaard, Nicklas Brustad, Jakob Stokholm, Klaus Bønnelykke, Hans Bisgaard, Bo Lund Chawes

**Affiliations:** COPSAC, Copenhagen Prospective Studies on Asthma in Childhood, Herlev and Gentofte Hospital, University of Copenhagen, 2820 Gentofte, Copenhagen, Denmark; ann-marie.schoos@dbac.dk (A.-M.M.S.); cevinther@gmail.com (C.V.); sarah.noergaard@dbac.dk (S.N.); nicklas.brustad@dbac.dk (N.B.); stokholm@copsac.com (J.S.); kb@copsac.com (K.B.); chawes@copsac.com (B.L.C.)

**Keywords:** MeSH, 25(OH)-vitamin D, vitamin D, determinants, genetics, environmental factors, pregnancy, children

## Abstract

Vitamin D insufficiency has become a common health problem worldwide, particularly among pregnant women and young children. Therefore, we sought to identify environmental, dietary, and genetic determinants of serum 25(OH)-vitamin D (25(OH)D) levels during pregnancy and early childhood. 25(OH)D was measured in women at 24-weeks of gestation (*n* = 738) and one-week postpartum (*n* = 284) in the population-based Danish COPSAC_2010_ mother–child cohort; and in cord blood (*n* = 257) and age 4 years (*n* = 298) in children from the at-risk COPSAC_2000_ mother–child cohort. Environmental, dietary, and genetic variables were tested for association with 25(OH)D using linear regression analyses. After adjusting for season of blood sampling, determinants of lower 25(OH)D levels during pregnancy in the women were higher pre-pregnancy BMI, lower age at birth, lower genetic vitamin D score, lower dietary vitamin D intake, and lower social circumstances. In children, the determinants were lower maternal age at birth, higher pre-pregnancy BMI, lower genetic vitamin D score, older siblings, exposure to tobacco smoking, and female sex. Genetics was an important determinant at all time points, alone explaining 2%–11% of the variance in 25(OH)D. Important determinants of circulating 25(OH)D levels during pregnancy and early childhood include environmental factors, diet, and to a large extent genetics.

## 1. Introduction

Humans get vitamin D from exposure to sunlight, diet, and dietary supplements. Dietary sources of vitamin D include oily fish, egg yolk, and fortified foods including milk, cheese, orange juice, infant milk formula, etc. [1]. Vitamin D deficiency affects bone mineralization and can lead to rickets, increased fracture risk, and osteoporosis [2], but has also been shown to associate with several non-communicable diseases including childhood asthma [3], diabetes [4], hypertension [5], cardiovascular events [6,7], obesity [8], cancer [9,10,11,12,13,14], and multiple sclerosis [15].

Circulating 25(OH)-vitamin D (25(OH)D) levels vary among humans, and according to the majority of single-center studies, vitamin D deficiency (defined as 25(OH)D of less than 50 nmol/L (20 ng/mL) [16,17,18]) is widespread in children, pregnant women, and breastfeeding mothers [19,20,21,22]. It has been estimated that vitamin D insufficiency, defined as 25(OH)D between 50 and 75 nmol/L (20–30 ng/mL) [16,17,18], affects as many as one-half of otherwise healthy adults in developed countries and that one billion people worldwide have vitamin D deficiency or insufficiency [1].

Both dietary and environmental factors are speculated to play a role in the development of low 25(OH)D [1,14,20], and we have previously shown that season of birth affected cord blood 25(OH)D [23]. Further, the high heritability of vitamin D insufficiency suggests that genetic determinants may also be important, and genome-wide association studies have replicated four genetic markers influencing 25(OH)D levels, including GC (encoding vitamin D binding protein), CYP2R1 (encoding a C-25 hydroxylase converting vitamin D3 to an active receptor ligand), as well as DHCR7 and NADSYN1 (both involved in cholesterol synthesis) [24,25,26,27]. The aim of this study was to identify environmental, dietary, and genetic determinants of vitamin D deficiency during pregnancy in the population-based Copenhagen Prospective Studies on Asthma in Childhood (COPSAC_2010_) mother–child cohort and during early childhood in the at-risk COPSAC_2000_ cohort.

## 2. Methods

### 2.1. Ethics

The Copenhagen Ethics Committee (HKF 01-289/96; H-B-2008-093) and The Danish Data Protection Agency (2008-41-1754; 2015-41-3696) approved the study. Oral and written informed consent was obtained from all parents at enrolment.

### 2.2. Study Populations

COPSAC_2000_ is a prospective at-risk birth cohort study of 411 children born to mothers with a history of asthma during 1998–2001 [28,29,30]. The children were enrolled at age 1 month and subsequently attended 6-monthly visits to the COPSAC research unit until age 3 and yearly thereafter until age 7.

COPSAC_2010_ is a prospective population-based mother–child cohort of 743 mothers and their 700 children recruited during 2009–2010 [31,32,33]. The women were included during pregnancy week 24. Exclusion criteria were gestational age above week 26, intake of vitamin D exceeding 600 IU/d, or any endocrine/heart/kidney disorders. Of the included women, 623 were enrolled in a double-blind randomized clinical trial of 2400 IU/d cholecalciferol D3 supplementation or placebo (Camette, Denmark A/S) from pregnancy week 24 to one week postpartum. All participants were instructed to continue the recommended supplementation of 400 IU/d cholecalciferol D3.

### 2.3. 25(OH)D Measurements

In COPSAC_2000_, serum 25(OH)D levels were measured at birth in cord blood and at 4 years of age. Cord blood was collected by needle puncture from the umbilical cord vein by midwives and subsequently sent to the COPSAC research unit. At age 4, a blood sample was collected from a peripheral vein at the COPSAC research unit. In COPSAC_2010_, maternal 25(OH)D levels were measured at week 24 of pregnancy and one week postpartum in blood samples collected at the COPSAC research unit. All the blood samples were centrifuged for 10 min at 4300 rpm to separate serum and subsequently frozen at −80°C until analysis, which was done by isotope dilution liquid chromatography-tandem mass spectrometry [34,35] at the Dept. of Clinical Biochemistry, Aarhus University Hospital, Denmark. Further details are given in the Appendix A.

### 2.4. Determinants

Anthropometry included child BMI at birth and age 4 years and maternal pre-pregnancy BMI. Length at birth was measured using an infantometer (Kiddimeter; Raven Equipment Ltd., Dunmow, Essex, England). Height at 4 years and maternal height was measured using a stadiometer (Harpenden; Holtain Ltd., Crymych, Dyfed, Wales).

Environmental determinants included maternal age at birth; season of birth; gestational age at birth; smoking during third pregnancy trimester (yes/no); maternal asthma status during third trimester (better, unchanged, worse) for COPSAC_2000_ and maternal asthma (yes/no) for COPSAC_2010_; mode of delivery (natural/Caesarean section); and social circumstances based on household income, maternal age, and maternal educational level. Postnatal exposures included season of blood sampling (winter (December–February), spring (March–May), summer (June–August), and fall (September–November)), older siblings (yes/no), and environmental tobacco exposure measured objectively as hair nicotine level (ng/mg) at age 1 year [36]. Race and gender were included for all participants.

Diet in COPSAC_2010_ included maternal mid-pregnancy dietary intake, obtained prospectively during a 4-week period by validated comprehensive food frequency questionnaires allowing for estimation of daily vitamin D intake [37]. The child’s dietary habit in COPSAC_2000_ was only available in the first year of life, and therefore not used in the analyses.

Genetics included genome-wide genotyping data of the mothers and children in both cohorts. In COPSAC_2000_, a vitamin D genetic score was composed of GC, CYP2R1, DHCR7, and NADSYN1 genotype data, where the score is the number of effect alleles previously associated with an increase in 25(OH)D [24,25,26]. In COPSAC_2010_, the vitamin D genetic score was composed of two GC single nucleotide polymorphisms (SNPs) genotypes (rs4588 and rs7041) as the other SNPs used in COPSAC_2000_ were not available in this cohort.

### 2.5. Statistics

Baseline characteristics are presented as median and interquartile range (IQR), mean and standard deviation (SD), or as number and percentage. Prior to analyses, the levels of 25(OH)D were adjusted for season by using a cosinus–sinus model (25(OH)D = sin (2 * π * day of blood sample/365.25) + cos (2 * π * day of blood sample/365.25)) as we wanted to examine determinants of 25(OH)D apart from the season of blood sampling. Thereafter, univariate linear regression models were used for crude analysis of the associations between the risk factors and 25(OH)D levels. Subsequently, multiple forward regression analyses were used as a variable selection method. The combination of variables that resulted in the lowest Akaike information criterion (AIC) determined the best model fit. Race was not included in the forward regression analysis, since genetic data were only obtained from Caucasians. Finally, multivariate regression analyses of all variables were performed. All statistical analyses were performed with RStudio version 3.3.0., considering a *p*-value < 0.05 as significant.

## 3. Results

### 3.1. Pregnant Women from COPSAC_2010_

A blood sample was available for 25(OH)D analysis at pregnancy week 24 in 738 (99%) of the 743 enrolled women and in 284 women at one week postpartum, who were randomized to placebo. The median (IQR) 25(OH)D level in the study group was 74.7 nmol/L (57–92) at pregnancy week 24 and 71 nmol/L (48–92) one week postpartum. At week 24 and one week postpartum, 14% and 22% of the women had deficient vitamin D levels (<50 nmol/L), respectively. Sufficient levels (>75 nmol/L) were measured in 52% of the women at week 24 and in 45% at one week postpartum. Baseline characteristics of the women are shown in Online Appendix A.

In Figure 1, maternal 25(OH)D levels at gestational week 24 and one week postpartum are depicted, stratified for season of blood sampling and categorized into sufficient (>75 nmol/L), insufficient (50–75 nmol/L), and deficient (<50 nmol/L). At week 24, 13% of the women had deficient 25(OH)D levels during summer compared to 17%, 10%, and 14% during winter, spring, and autumn, respectively. Sufficient levels were measured in approximately half of the women (40%–55%), with a slight preponderance during spring (55%). At one week postpartum, slightly more women (19%) had deficient 25(OH)D during summer compared to 21%, 21%, and 26% during winter, spring, and autumn, respectively. Sufficient levels were measured in 41%–54% of the women.

The absolute 25(OH)D levels varied significantly with season. For week 24 when using winter as reference, 25(OH)D was 19.3 nmol/L higher during summer (95% CI [14.2; 24.4], *p* < 0.001) and 10.7 nmol/L higher during autumn (95% CI [6.0; 15.4], *p* < 0.001). For one week postpartum, 25(OH)D was 27.2 nmol/L higher during summer vs. winter (95% CI [17.3; 37.2], *p* < 0.001), but no significant differences were found for the rest of the seasons vs. winter. The 25(OH)D levels in week 24 and one week postpartum were associated both in the unadjusted and season-adjusted analyses: 0.54 (95% CI [0.41; 0.67], *p* < 0.001) and 0.68 (95% CI [0.56; 0.80], *p* < 0.001), respectively. We found no difference between the 25(OH)D levels at week 24 and one week postpartum: mean difference unadjusted was −3.2 nmol/L (95% CI [−6.9; 0.4]) and adjusted was 0.4 nmol/L (95% CI [−2.7; 3.5]) (Figure 2).

### 3.2. Environmental, Dietary, and Genetic Determinants of Maternal 25(OH)D Levels

Univariate analyses of week 24 samples showed that 25(OH)D levels were positively correlated to Caucasian origin, increasing maternal age at birth, both vitamin D SNPs, and increasing dietary intake, and negatively correlated to increasing maternal pre-pregnancy BMI. At one week postpartum, 25(OH)D levels were positively correlated to higher social circumstances, both vitamin D SNPs, and negatively correlated to maternal pre-pregnancy BMI. We did not find any other significant correlations with maternal 25(OH)D levels (Table 1).

To examine the most important determinants of 25(OH)D levels, a multivariate forward regression analysis was made. The following variables were included in the analysis of week 24: maternal pre-pregnancy BMI, age at birth, smoking in third trimester, maternal asthma, social circumstances, diet, and the vitamin D SNPs (rs4588 and rs7041). The same variables were used for the one week postpartum analysis, but gestational age at birth and Caesarian section were also included. The final models included maternal pre-pregnancy BMI, age at birth, diet, and rs4588 for week 24, explaining 7% of the variance in 25(OH)D levels (R^2^ = 0.07, *p* < 0.001), and maternal pre-pregnancy BMI, social circumstances, and both vitamin D SNPs for one week postpartum, explaining 10.3% of the variance in 25(OH)D levels (R^2^ = 0.103, *p* < 0.001) (Table 2).

To further explore the effect of the different determinants on 25(OH)D levels, a multivariate regression analysis was performed. At week 24 and one week postpartum, the environmental determinants explained 2.3% (R^2^ = 0.023, *p* = 0.002) and 1.3% (R^2^ = 0.013, *p* = 0.2) of the variance in 25(OH)D, genetics explained 3.5% (R^2^ = 0.029, *p* < 0.001) and 5.4% (R^2^ = 0.054, *p* < 0.001), and diet explained 1.6% (R^2^ = 0.016, *p* = 0.001) and 0.2% (R^2^ = 0.002, *p* = 0.24), respectively.

### 3.3. Children from COPSAC_2000_

Cord blood was available for 25(OH)D analysis in 257 (63%) of the 411 children and 298 (73%) had 25(OH)D measured at age 4 years. The median (IQR) 25(OH)D cord blood level was 40.9 nmol/L (28–55) and age 4 years was 75 nmol/L (59–91). In cord blood, 67% of the children had deficient vitamin D levels and only 8% had sufficient levels. At age 4 years, 14% had deficient and 50% had sufficient levels. Baseline characteristics of the children are given in Online Appendix A.

Fewer infants with deficient 25(OH)D cord blood levels were born during summer (51%) compared to 77%, 75%, and 70% born during winter, spring, and autumn (Figure 3). Only 4%–9% had sufficient levels at all seasons, with a slight preponderance of children born during summer (9%). At age 4 years, fewer children (3%) had deficient 25(OH)D during summer compared to 17%, 17%, and 4% during winter, spring, and autumn. Sufficient levels were measured in 36%–70% of the children.

When using winter as reference, cord blood levels of 25(OH)D were 12.3 nmol/L higher in children born during summer (95% CI [5.5; 19.3], *p* < 0.001). For age 4 years, the levels measured during summer and autumn were 19.1 nmol/L (95% CI [11.1; 27.1], *p* < 0.001) and 12.4 nmol/L (95% CI [4.8; 20.1], *p* = 0.002) higher, respectively. No other seasonal differences were observed (data not shown). The 25(OH)D levels in cord blood and by age 4 years were associated both in the unadjusted and season-adjusted analyses: 0.18 (95% CI [0.02; 0.34], *p* = 0.03) and 0.25 (95% CI [0.08; 0.42], *p* = 0.004), respectively. We found an increase in the 25(OH)D levels from cord blood to age 4 years: mean difference unadjusted: 35.5 nmol/L (95% CI [31.7; 39.4]) and adjusted: 34.7 nmol/L (95% CI [31.0; 38.4]) (Figure 4).

### 3.4. Environmental, Dietary, and Genetic Determinants of Child 25(OH)D Levels

In the univariate analyses, cord blood 25(OH)D level was positively correlated to maternal age at birth, the vitamin D genetic score, and negatively correlated to smoking during the third trimester. It was borderline positively correlated to Caucasian origin and social circumstances. At age 4, 25(OH)D was positively correlated to the vitamin D genetic score, male gender, and negatively correlated to tobacco exposure (Table 3).

A forward multivariate regression analysis for cord blood 25(OH)D yielded a final model including maternal age at birth, smoking during third trimester, older siblings, and the vitamin D genetic score, explaining 9.6% (R^2^ = 0.096, *p* < 0.001) of the variance in 25(OH)D. Particularly, having older siblings resulted in 8.7 nmol/L lower levels (*p* = 0.001) and smoking during the third trimester resulted in 6.5 nmol/L lower levels (*p* = 0.07) (Table 4). The final forward regression model for 4 years included the vitamin D genetic score, child BMI, and gender, explaining 10.8% (R^2^ = 0.108, *p* < 0.001) of the variance. Male gender resulted in 6.7 nmol/L higher 25(OH)D (*p* = 0.01).

The environmental factors explained 5.2% (R^2^ = 0.052, *p* = 0.016) of the variance in 25(OH)D levels and genetics 2.9% (R^2^ = 0.029, *p* = 0.006). At 4 years, the environmental factors explained 1.8% (R^2^ = 0.018, *p* = 0.105) and genetics 10.4% (R^2^ = 0.104, *p* < 0.001).

## 4. Discussion

### 4.1. Primary Findings

In both pregnant women from COPSAC_2010_ and children from COPSAC_2000_, we found a high prevalence of vitamin D deficiency and insufficiency despite the fact that all Danish pregnant women and children up to 2 years of age are advised to take a vitamin D supplement of 400 IU/d [38]. Less than half of the women had sufficient levels at pregnancy week 24 and one week postpartum, just 7% of the children had sufficient cord blood levels, and less than half had sufficient levels at age 4 years. The best determinants of season-adjusted 25(OH)D levels in pregnant women included pre-pregnancy BMI, social circumstances, age at birth, diet, and vitamin D genetics, whereas the best determinants in young children were maternal age at birth, older siblings, smoke exposure, child BMI, gender, and vitamin D genetics, with genetics alone explaining 2%–11% of the variance in 25(OH)D levels in the cohorts. These findings suggest that environment, diet, and genetics all influence 25(OH)D levels in pregnant women and young children, which is important for vitamin D supplementation strategies.

### 4.2. Strengths and Limitations

The main strength of this study is the single-center set-up with thorough longitudinal clinical phenotyping and data collection in both the COPSAC_2000_ and the COPSAC_2010_ cohort. All assessments were solely performed by the COPSAC research pediatricians, who ensured consistency in procedures, definitions of conditions, and data capture methods, and thereby limited inter-observer variation.

Another strength is the objective assessment of 25(OH)D levels by state-of-the-art LC-MS two times during pregnancy, in cord blood, and at age 4 years. Even though cord blood 25(OH)D level primarily reflects exposure during the third trimester and may be contaminated by maternal blood, it is considered a more accurate measure of fetal exposure compared to calculations from questionnaires on maternal diet, since only 10% of vitamin D is obtained through diet [39].

The comprehensive assessment of environmental exposure, diet, and genetics is another strength providing many relevant 25(OH)D determinants for analysis, but we lack information on certain variables, including the use of sunscreen or protective clothing, outdoor physical activity, and diet information in COPSAC_2000_. Regarding skin pigmentation, 96%–97% in the study groups were of Caucasian origin and we do not expect this to be of great influence; therefore, analyses were not adjusted for ethnicity. We observed that season had a major influence on 25(OH)D levels, but since it reflects the time of blood sampling rather than a long-term determinant of 25(OH)D level we adjusted all the risk-factor analyses for this. Regarding genetics, we chose a simple and clinically applicable approach using four genes where we had available genotype information, which have been replicated across genome-wide association studies to contribute to vitamin D levels. However, there are likely to be far more gene variants associated both with increased and decreased vitamin D, but these were not available in our study.

The population-based COPSAC_2010_ cohort provides findings generalizable to other populations, and it would have been interesting to analyze 25(OH)D levels in the children at age 4 years to compare them to the general population. Unfortunately, we only measured 25(OH)D levels in the children in the COPSAC_2000_ cohort who were born to asthmatic mothers and thus not directly comparable to the general population. However, this should not affect the ability to analyze determinants of 25(OH)D levels within the cohort, and some of the findings are in line with a similar study of unselected children [40].

### 4.3. Interpretation

We found a high prevalence of vitamin D deficiency and insufficiency in both mothers and children. After adjusting for season of blood sampling, significant determinants of lower 25(OH)D levels during pregnancy were higher pre-pregnancy BMI, lower age at birth, lower social circumstances, lower dietary intake of vitamin D, and lower genetic vitamin D score. In the children, the determinants were lower maternal age at birth, older siblings, exposure to tobacco smoking, higher pre-pregnancy BMI, female gender, and lower genetic score. Genetics alone contributed to 2%–11% of the variance in 25(OH)D levels in the pregnant women and young children.

Our finding of a high prevalence of vitamin D deficiency is in line with other studies of Danish pregnant women [39] and children aged 3 months to 6 years [40]. Season of blood sampling played a major role, as we found that 25% fewer children had deficient cord blood levels if they were born during summer compared to winter and spring, which aligns with cord blood results reported from an Australian prospective birth cohort study [41]. In the COPSAC_2000_ cohort, the season variation (summer vs. winter) of 25(OH)D levels in children was greater at age 4 years than in cord blood measurements, which is probably due to a greater variation of outside activity among children compared with mothers’ activity in the third trimester reflected in the cord blood levels. In the COPSAC_2010_ cohort, the seasonal variation of being vitamin D deficient between week 24 and one week post-partum was less profound, and this could be due to increased skin coverage and less outdoor activity in late pregnancy.

We found that 25(OH)D levels increased during childhood from birth until age 4, which is in line with results from another birth cohort study [42], but opposite to findings from a study of healthy children observing a decrease in 25(OH)D from 3 months to 6 years [40]. An increase in vitamin D levels could be explained by different adherence to supplementation strategies or the fact that we measured 25(OH)D in cord blood, where levels are generally lower than later in childhood. Exposure to sunlight during outdoor activities is presumably the main environmental driver of the increase from birth till age 4 years, but our data also suggest that dietary and lifestyle habits associated with BMI may be of importance.

In our study, we observed associated but stable 25(OH)D levels from week 24 to one week postpartum. Previous longitudinal studies have reported both a decrease in 25(OH)D concentrations throughout pregnancy and in the period after delivery [43,44] and an increase through the first, second, and third trimesters [45]. We speculate that our findings could be a result of increased adherence to the recommended supplementation guidelines, as the mothers were enrolled in a clinical trial.

We focused on vitamin D status during pregnancy and early childhood, whereas many previous studies have tried to estimate determinants of long-term vitamin D status in adulthood [46,47,48,49,50]. That may be more relevant as a model for assessing the risk of developing chronic diseases such as cancers, inflammatory diseases, and osteoporosis, while our model is a better predictor of vitamin D deficiency in shorter vulnerable periods where personalized vitamin D supplementation strategies would be particularly beneficial.

We observed higher 25(OH)D in children of male sex and in children and women with lower BMI. This could be explained by males and subjects with lower BMI being more outside and more physically active, but also by the fact that vitamin D is fat-soluble a therefore deposited in fat tissue in girls and overweight persons. Tobacco exposure was also associated with lower child 25(OH)D level, which could be caused by other lifestyle factors found more prevalent among smokers, such as less physical activity and nutritional deficiencies. A more direct effect of tobacco exposure could include an increased hepatic metabolism of 25(OH)D, as smokers have been shown to have enhanced hepatic degradation of estrogen [51]. In both the pregnant women and in the children, we found higher 25(OH)D levels with higher maternal age at birth, higher social circumstances, and in persons of Caucasian origin, which is in line with several previous studies [40,46,47,52,53]. The association with higher maternal age and higher social circumstances may once again be due to lifestyle factors, including higher dietary intake of vitamin D containing foods and higher adherence to the recommended daily vitamin D supplement.

In the forward regression analysis, smoke exposure and older siblings were the variables with the greatest effect on cord blood 25(OH)D when looking at the estimates. Correspondingly, at age 4 years being male had the most influence on 25(OH)D levels. However, when looking at all the environmental factors they only accounted for 2%–5% of the variance in 25(OH)D levels, with the greatest effect on cord blood levels. This may be due to our adjustment for seasons prior to the risk-factor analysis or due to other unknown determinants lacking in our model (e.g., outdoor physical activity).

Regarding heritability, both the pregnant women and young children had significantly higher 25(OH)D levels with higher vitamin D genetic score, that is, increasing number of wildtype alleles involved in the vitamin D metabolism, explaining in itself 2%–11% of the variance in 25(OH)D. In the forward regression analysis, genetics was included in all the final models from both pregnant women and children and was the variable with greatest influence on 25(OH)D in the pregnant women, when looking at the estimates. Undoubtedly, the genetic regulation of the vitamin D pathway and levels (i.e., both increased and decreased levels) is very complex and presumably not yet completely understood. We chose a simple and clinically applicable approach using four genes where we had available genotype information, but there are likely to be far more gene variants associated with both increased and decreased vitamin D [54]. In a future study, it would be interesting to explore a wider range of gene variants, gene expression data, as well as gene–gene and gene–environment interactions contributing to trajectories of vitamin D levels during pregnancy and early childhood.

In the pregnant women, diet was significantly associated with 25(OH) levels at gestational week 24, where the food frequency questionnaire was fulfilled. Diet was not a determinant of 25(OH)D at one week postpartum, which aligns with one previous study [53] but is in contrast to most other studies [46,49,50,52], and could be explained by changing dietary habits in pregnancy. Diet was included in the week-24 forward regression model but explained <2% of the variance in 25(OH)D.

Overall, our regression models only explained 7%–11% of the variance in 25(OH)D status in pregnant women and young children, which is lower than the 21%–40% reported in previous studies of pregnant women, adolescents, and adults [47,48,52]. This is probably because we adjusted the 25(OH)D levels for season of blood sampling to minimize this as a confounder for other risk factors, particularly when assessing determinants of 25(OH)D at several time points in the same subject [55].

## 5. Conclusions

We found a high prevalence of vitamin D deficiency during pregnancy and in early childhood, and that important determinants include environmental factors, diet, and vitamin D genetics. These findings may be used to identify at-risk individuals providing a more personalized approach to vitamin D supplementation advice for pregnant women and young children.

## Figures and Tables

**Figure 1 children-06-00116-f001:**
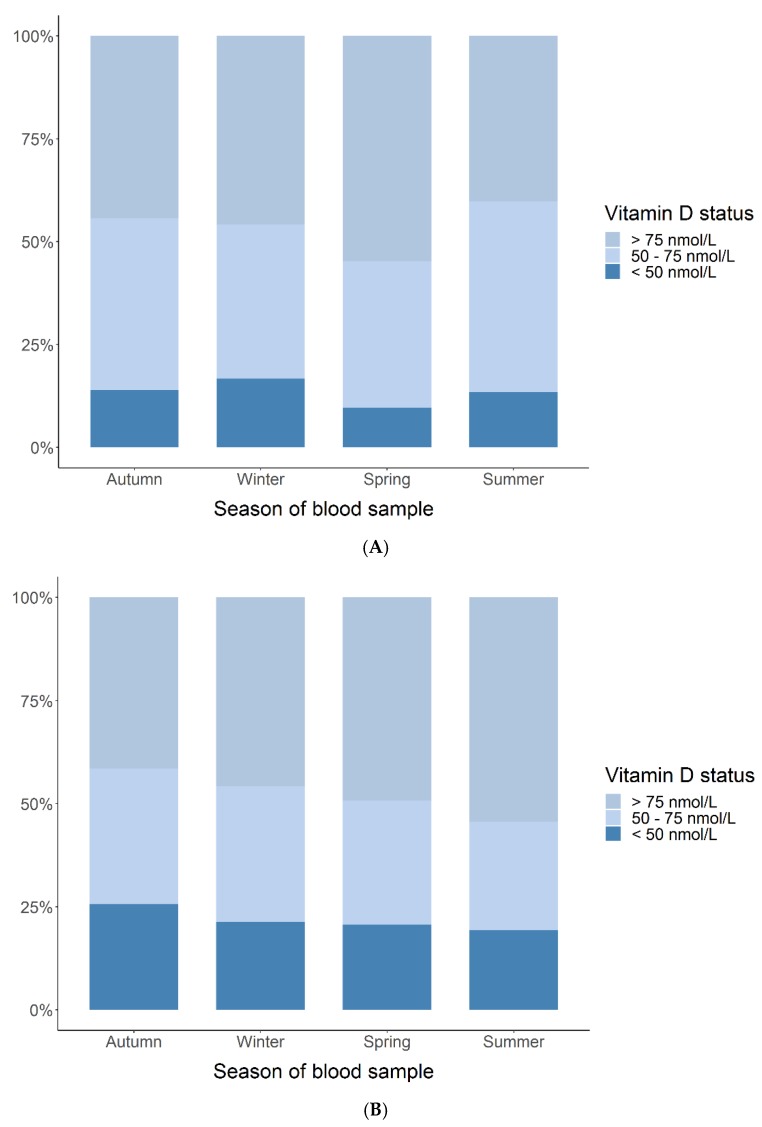
Maternal 25(OH)-vitamin D (25(OH)D) status (deficient (<50 nmol/L), insufficient (50–75 nmol/L), sufficient (>75 nmol/L)) at week 24 (**A**) and one week postpartum (**B**) stratified for season of blood sampling.

**Figure 2 children-06-00116-f002:**
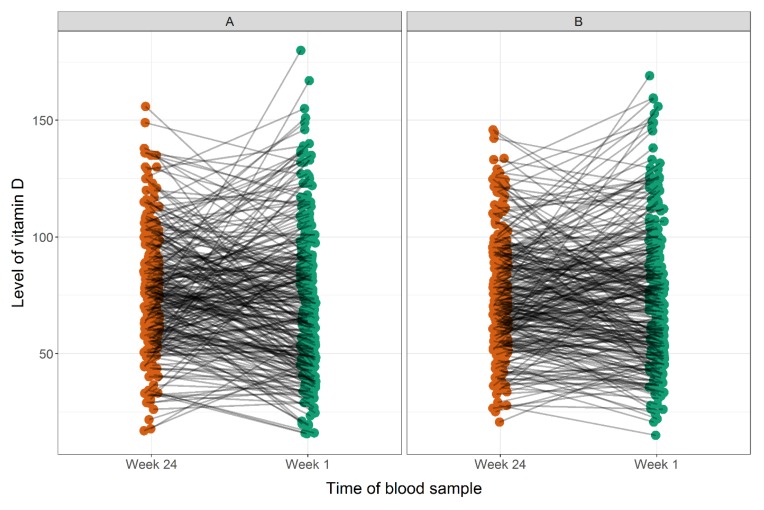
Correlation between maternal 25(OH)D level measured at week 24 and one week postpartum in nmol/L unadjusted (**A**) and adjusted for season of blood sampling (**B**).

**Figure 3 children-06-00116-f003:**
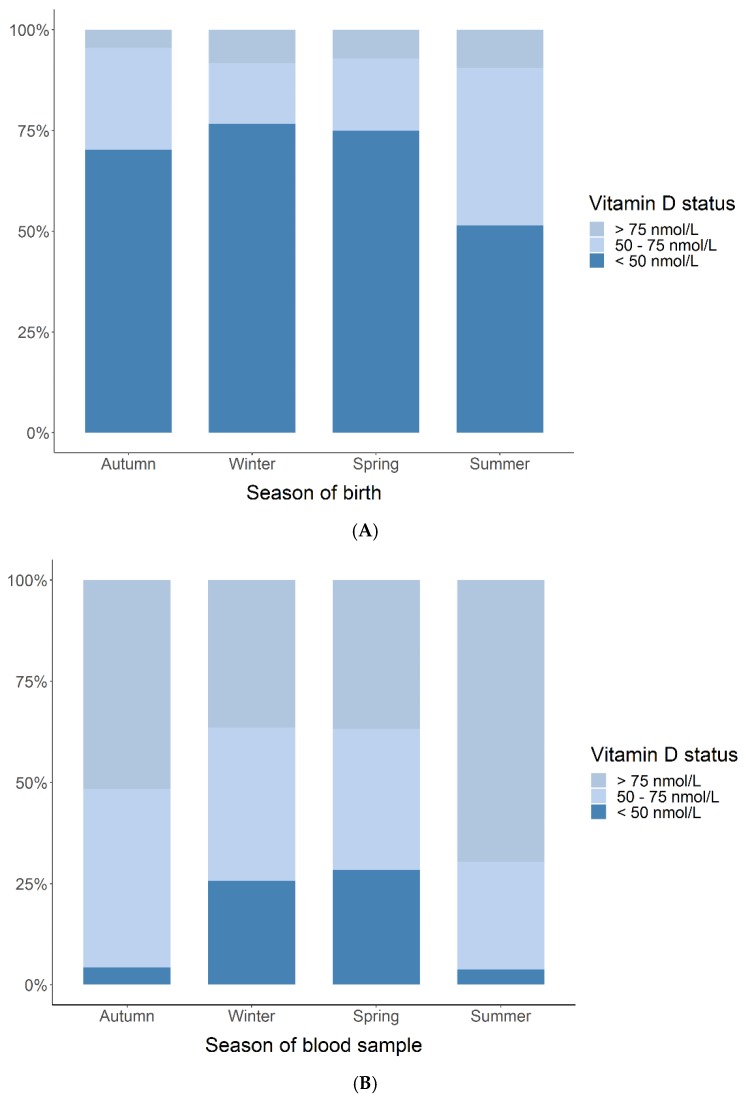
Child 25(OH)D status (deficient (<50 nmol/L), insufficient (50–75 nmol/L), sufficient (>75 nmol/L)) measured in cord blood (**A**) and at 4 years (**B**) stratified for season of blood sampling.

**Figure 4 children-06-00116-f004:**
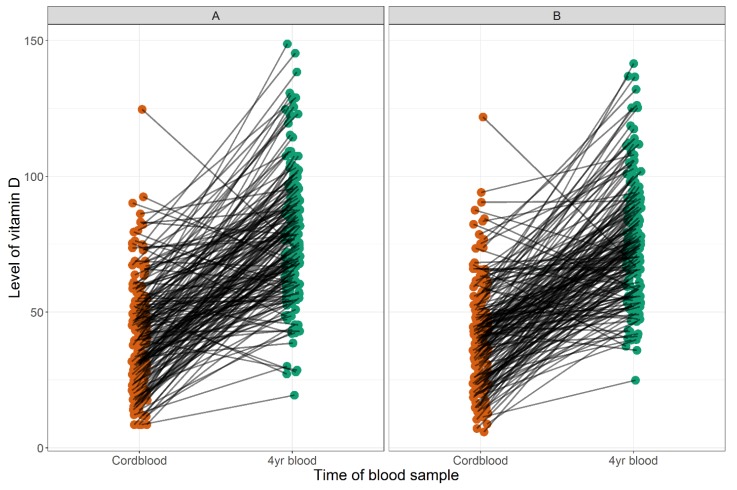
Correlation between cord blood and age 4 years 25(OH)D level in nmol/L unadjusted (**A**) and adjusted for season of blood sampling (**B**).

**Table 1 children-06-00116-t001:** Univariate analyses of determinants for maternal season-adjusted 25(OH)D levels during pregnancy.

	Week 24 Gestation β-Coefficient (nmol/L)	95% CI	*p*-Value	One Week Postpartum β-Coefficient (nmol/L)	5% CI	*p*-Value
Caucasian vs. non-Caucasian	11.71	[3.00; 20.4]	0.008	15.55	[−1.09; 32.2]	0.07
Maternal BMI, kg/m²	−0.74	[−1.17; −0.30]	0.001	−1.09	[−1.92; −0.26]	0.01
Gestational age, weeks	-	-	-	−1.40	[−3.60; 0.80]	0.21
Caesarian section vs. vaginal delivery	-	-	-	0.01	[−8.61; 8.63]	1.00
Maternal age at birth, years	0.41	[0.02; 0.80]	0.038	0.41	[−0.41; 1.22]	0.33
SNP rs4588	7.58	[4.81; 10.4]	<0.001	12.49	[6.76; 18.2]	<0.001
SNP rs7041	4.74	[2.26; 7.23]	<0.001	7.83	[2.90; 12.8]	0.002
Maternal asthma, yes vs. no	−2.93	[−6.96; 1.10]	0.15	−0.19	[−8.11; 7.73]	0.96
Maternal smoking in third trimester, yes vs. no	−1.06	[−10.6; 8.50]	0.83	−11.78	[−27.9; 4.32]	0.15
Social circumstances, PCA score	1.21	[−0.57; 2.98]	0.18	4.59	[0.89; 8.29]	0.015
Dietary vitamin D intake, μg/day	1.16	[0.47; 1.84]	0.001	0.88	[−0.60; 2.35]	0.24

**Table 2 children-06-00116-t002:** Results from final multivariate forward regression analyses of maternal season-adjusted 25(OH)D levels.

	Week 24 Gestation β-Coefficient (nmol/L)	95% CI	*p*-Value	One Week Postpartum β-Coefficient (nmol/L)	95% CI	*p*-Value
Maternal BMI, kg/m²	−0.70	[−1.20; −0.25]	0.002	−1.05	[−1.50; −0.15]	0.02
Maternal age at birth, years	0.40	[−0.08; 0.87]	0.10	-	-	-
rs4588	8.30	[5.00; 11.60]	<0.001	7.50	[1.10; 18.50]	0.12
rs7041	-	-	-	5.80	[−4.80; 10.10]	0.15
Social circumstances, PCA score	-	-	-	3.70	[0.60; 8.29]	0.09
Dietary vitamin D intake, μg/day	1.03	[0.31; 1.76]	0.005	-	-	-

**Table 3 children-06-00116-t003:** Univariate analyses of determinants for cord blood and age 4 years season-adjusted 25(OH)D levels.

	Cord Blood β-Coefficient (nmol/L)	95% CI	*p*-Value	4 Years β-Coefficient (nmol/L)	95% CI	*p*-Value
Male	−0.47	[−5.38; 4.43]	0.85	6.39	[1.06; 11.7]	0.019
Caucasian	13.5	[−0.51; 27.5]	0.059	4.72	[−10.2;19.6]	0.53
BMI, kg/m²	−0.19	[−2.08; 1.69]	0.84	−1.58	[−3.92; 0.75]	0.18
Gestational age, weeks	0.61	[−0.99; 2.21]	0.46	-	-	-
Caesarian section	−0.18	[−6.36; 6.01]	0.96	-	-	-
Maternal age at birth, years	0.73	[0.16; 1.30]	0.012	-	-	-
Genetic score	2.00	[0.58; 3.44]	0.006	4.45	[2.90; 6.01]	<0.001
Older siblings at birth	−4.01	[−8.95; 0.94]	0.11	-	-	-
Maternal smoking in third trimester	−8.04	[−14.7; −1.43]	0.017	-	-	-
Nicotine in hair, ng/mg	-	-	-	−0.03	[−0.05; 0.00]	0.021
Asthma status in third trimester						
Better	1.86	[−3.81; 7.53]	0.52	-	-	-
Unchanged	Ref.					
Worse	−1.47	[−7.90; 4.95]	0.65	-	-	-
Social circumstances, PCA score	2.59	[−0.04; 5.21]	0.053	1.54	[−1.30; 4.37]	0.29

**Table 4 children-06-00116-t004:** Results from final multivariate forward regression analyses of child season-adjusted 25(OH)D levels.

	Cord Blood β-Coefficient (nmol/L)	95% CI	*p*-Value	4 Years β-Coefficient (nmol/L)	95% CI	*p*-Value
Maternal age at birth, years	0.9	[0.26; 1.54]	0.006	-	-	-
Genetic score	2.07	[0.7; 3.46]	0.003	4.00	[2.46; 5.56]	<0.001
Older siblings at birth	−8.71	[−14.2; −3.24]	0.001	-	-	-
Maternal smoking in third trimester	−6.50	[−13.6; 0.52]	0.07	-	-	-
Child BMI, kg/m^2^	-	-	-	−1.78	[−4.03; 0.46]	0.12
Male gender	-	-	-	6.7	[1.54; 11.93]	0.01

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
