# Peer review of "Environmental and Genetic Determinants of Serum 25(OH)-Vitamin D Levels during Pregnancy and Early Childhood"

_children, 2019, doi:10.3390/children6100116_

Round 1

Reviewer 1 Report

A well conducted and interesting study.

In particular the targeted (but limited) genetic findings here have important implications for supplementation strategies. At 4 years of age variance in 4 genotypes explained 10% variance. This seems very high given previous 25OHD GWAS studies explaining less although perhaps this was due to a more varied population).

For instance two additional loci harbouring genome-wide significant variants (P = 4.7×10−9 at rs8018720 in SEC23A, and P = 1.9×10−14 at rs10745742 in AMDHD1) were recently established Jiang et al., Nature Comms 2018. One wonders how much more variance would be explained if these variants were considered in this homogenous population?

I think Figs 3 and 4 are the most remarkable data in this submission. Here clearly environment has to dictate the overall dramatic increase in 25OHD status. Could the authors comment?

Strengths

The analytical component uses gold standard methods and external calibration.

The sample includes matched samples from either mothers or offspring longitudinally which is rare and novel.

Good consideration of seasonality of sampling and de-seasonalising of data.

A number of appropriate co-variates were examined.

Weaknesses

Sample size is modest and possibly underpowered for genetic conclusions made

Only 4 genes previously associated +with increased 25OHD were examined when there are likely to be far more gene variants controlling D levels and possibly also affected by environment.

Also there are well known variants associated with decreased 25OHD i.e Cyp 24A1 (Jiang et al.,). One wonders why such variants were not also examined. This certainly would have strengthened some of the conclusions on functional links between genotype and 25OHD levels.

Author Response

Response to reviewer 1:

A well conducted and interesting study.

In particular the targeted (but limited) genetic findings here have important implications for supplementation strategies. At 4 years of age variance in 4 genotypes explained 10% variance. This seems very high given previous 25OHD GWAS studies explaining less although perhaps this was due to a more varied population).

For instance two additional loci harbouring genome-wide significant variants (P = 4.7×10−9 at rs8018720 in SEC23A, and P = 1.9×10−14 at rs10745742 in AMDHD1) were recently established Jiang et al., Nature Comms 2018. One wonders how much more variance would be explained if these variants were considered in this homogenous population? 

Comment 1: I think Figs 3 and 4 are the most remarkable data in this submission. Here clearly environment has to dictate the overall dramatic increase in 25OHD status. Could the authors comment?

Response 1: As we state i lines 272-274, we believe that the increase in vitamin D levels could be explained by different adherence to supplementation strategies or the fact that we measured 25(OH)D in cord blood, where levels are generally lower than later in childhood. We added the following sentence in lines 278-280:

Exposure to sunlight during outdoor activities is presumably the main environmental driver of the increase from birth till age 4 years, but our data also suggest that dietary and lifestyle habits associated with BMI may be of importance.”

Comment 2: Strengths

The analytical component uses gold standard methods and external calibration.

The sample includes matched samples from either mothers or offspring longitudinally which is rare and novel.

Good consideration of seasonality of sampling and de-seasonalising of data.

A number of appropriate co-variates were examined.

Response 2: thank you.

Weaknesses

Sample size is modest and possibly underpowered for genetic conclusions made

Comment 3: Only 4 genes previously associated +with increased 25OHD were examined when there are likely to be far more gene variants controlling D levels and possibly also affected by environment. Also there are well known variants associated with decreased 25OHD i.e Cyp 24A1 (Jiang et al.,). One wonders why such variants were not also examined. This certainly would have strengthened some of the conclusions on functional links between genotype and 25OHD levels.

Response 3: Undoubtedly, the genetic regulation of the vitamin D pathway and levels; i.e. both increased and decreased levels is very complex and presumably not yet completely understood. We chose a simple and clinically applicable approach using 4 genes where we had available genotype information, which have been replicated across GWAS studies to contribute to vitamin D levels. We agree that in a future study it would be interesting to explore a wider range of gene variants, gene expression data, gene-gene and gene-environment interactions contributing to trajectories of vitamin D levels during pregnancy and early childhood. We have now added a paragraph on this weakness of our study in lines 243-247:

Regarding genetics, we chose a simple and clinically applicable approach using 4 genes where we had available genotype information, which have been replicated across GWAS studies to contribute to vitamin D levels. However, there are likely to be far more gene variants associated both with increased and decreased vitamin D, but these were not available in our study.”

We also added a paragraph in the discussion on lines 331-338 and referenced the study by Jiang and coworkers:

“Undoubtedly, the genetic regulation of the vitamin D pathway and levels, i.e. both increased and decreased levels, is very complex and presumably not yet completely understood. We chose a simple and clinically applicable approach using 4 genes where we had available genotype information, however, there are likely to be far more gene variants associated both with increased and decreased vitamin D[54]. In a future study it would be interesting to explore a wider range of gene variants, gene expression data, gene-gene and gene-environment interactions contributing to trajectories of vitamin D levels during pregnancy and early childhood.”

Reviewer 2 Report

This is a well-organized and well-presented study. I think that the genetics adds relevance as more and more information is gathered about pharmacogenomics. 

Editorial suggestion: lines 38-40: given broad audience, could also include ng/ml. E.g. 50 nmol/L (20 ng/ml), etc.  Perhaps a more UTD reference than ref.16,17 re: levels of vitamin D deficiency/insufficiency, e.g.  https://www.ncbi.nlm.nih.gov/pmc/articles/PMC2902062/ -which references WHO definitions, and/or others.    Strengths of study: large numbers with appropriate statistical analysis and contribution of vitamin D  levels in specific populations addressed.    The genetic studies are most relevant and in the age of pharmacogenomics may have important implications in the future.    Limitation: should have an explanation why there are no age 4 pediatric levels reported from COPSAC 2010 (line 229)- the state of vitamin D sufficiency in a large "generalizable population" as the authors refer to it, would have been valuable to know, regardless of how they chose to compare the results (or not) in the study. Also, to know how the levels in these children compared to those the authors reference in ref.#39.   

Author Response

Response to reviewer 2:

Comment 1: This is a well-organized and well-presented study. I think that the genetics adds relevance as more and more information is gathered about pharmacogenomics. 

Response 1: thank you

Editorial suggestion:

Comment 2: lines 38-40: given broad audience, could also include ng/ml. E.g. 50 nmol/L (20 ng/ml), etc. 

Response 2: This has now been added to lines 38-40.

Comment 3: Perhaps a more UTD reference than ref.16,17 re: levels of vitamin D deficiency/insufficiency, e.g.  https://www.ncbi.nlm.nih.gov/pmc/articles/PMC2902062/ -which references WHO definitions, and/or others.   

Response 3: This reference has now been added in lines 39 and 41.

Comment 4: Strengths of study: large numbers with appropriate statistical analysis and contribution of vitamin D  levels in specific populations addressed.    The genetic studies are most relevant and in the age of pharmacogenomics may have important implications in the future.   

Response 4: thank you.

Comment 5: Limitation: should have an explanation why there are no age 4 pediatric levels reported from COPSAC 2010 (line 229)- the state of vitamin D sufficiency in a large "generalizable population" as the authors refer to it, would have been valuable to know, regardless of how they chose to compare the results (or not) in the study. Also, to know how the levels in these children compared to those the authors reference in ref.#39.   

Response 5: This would have been very interesting, but we have not measured the 25(OH)D levels in the offspring in COPSAC2010. This has been added as a limitation in the discussion section lines 244-251:

“The population-based COPSAC2010 cohort provides findings generalizable to other populations, and it would have been interesting to analyze 25(OH)D levels in the children at age 4 years to compare them to the general population. Unfortunately, we only measured 25(OH)D levels in the children in the COPSAC2000 cohort who are born to asthmatic mothers and thus not directly comparable to the general population.”